# Zebrafish Models for Skeletal Muscle Senescence: Lessons from Cell Cultures and Rodent Models

**DOI:** 10.3390/molecules27238625

**Published:** 2022-12-06

**Authors:** Shogo Ichii, Izumi Matsuoka, Fumiyoshi Okazaki, Yasuhito Shimada

**Affiliations:** 1Graduate School of Bioresources, Mie University, Tsu, Mie 514-8507, Japan; 2Graduate School of Regional Innovation Studies, Mie University, Tsu, Mie 514-8507, Japan; 3Zebrafish Drug Screening Center, Mie University, Tsu, Mie 514-8507, Japan; 4Department of Bioinformatics, Mie University Advanced Science Research Promotion Center, Tsu, Mie 514-8507, Japan; 5Department of Integrative Pharmacology, Mie University Graduate School of Medicine, Tsu, Mie 514-8507, Japan

**Keywords:** sarcopenia, drug screening, animal models, skeletal muscle enlargement

## Abstract

Human life expectancy has markedly increased over the past hundred years. Consequently, the percentage of elderly people is increasing. Aging and sarcopenic changes in skeletal muscles not only reduce locomotor activities in elderly people but also increase the chance of trauma, such as bone fractures, and the incidence of other diseases, such as metabolic syndrome, due to reduced physical activity. Exercise therapy is currently the only treatment and prevention approach for skeletal muscle aging. In this review, we aimed to summarize the strategies for modeling skeletal muscle senescence in cell cultures and rodents and provide future perspectives based on zebrafish models. In cell cultures, in addition to myoblast proliferation and myotube differentiation, senescence induction into differentiated myotubes is also promising. In rodents, several models have been reported that reflect the skeletal muscle aging phenotype or parts of it, including the accelerated aging models. Although there are fewer models of skeletal muscle aging in zebrafish than in mice, various models have been reported in recent years with the development of CRISPR/Cas9 technology, and further advancements in the field using zebrafish models are expected in the future.

## 1. Introduction

In recent years, the average global life expectancy has been greater than 70 years. In fact, many developed countries have an average life expectancy of 80 years. For example, the average life expectancy at birth is 78.5, 81.4, 81.7, 82.5, 83.1, 82.2, and 84.3 years in the US, UK, Germany, France, Italy, Canada, and Japan, respectively (World Health Organization, 2020). To lead a prosperous and independent life and enjoy sports and travel after retirement, activities of daily living (ADL) and daily self-care activities, such as mobility and eating, must be maintained. With aging, our ADLs trend downward, which serves as one of the reasons for the decline in muscle mass. The skeletal muscle, the largest organ in the human body, plays an indispensable role in maintaining health as it functions as a metabolic organ and is responsible for movement and physical activity. According to epidemiological studies, people who maintain skeletal muscle mass are less likely to become sick and tend to live longer [1,2]. Therefore, the maintenance of skeletal muscles is key to a super-aging society.

Elderly people are at an increased risk for the following two skeletal muscle diseases: disuse muscular atrophy and sarcopenia. Disuse muscle atrophy is caused by prolonged rest due to severe injury, surgery, or hospitalization [3]. In atrophic muscles, the amount of DNA does not change; however, the amount of RNA is markedly reduced, suggesting that the amount of protein is decreased in atrophied muscles [4]. One of the decreases in RNA synthesis is caused by the degradation of signaling molecules in the IGF-1-mediated protein synthesis pathway [5]. Another muscle-related disease in the elderly population, sarcopenia, is defined as a loss of muscle mass [6]; however, recent definitions of sarcopenia include loss of muscle strength and muscle function [7,8,9,10,11,12,13]. Sarcopenia contributes to frailty, resulting in reduced ADL and quality of life [7,14]. Sarcopenia is classified as either primary or secondary. Primary sarcopenia is caused by muscle mass loss due to aging, while secondary sarcopenia is caused by muscle mass loss due to activity, disease, and nutrition [7]. Strictly speaking, disuse muscular atrophy differs from sarcopenia as the number of cells does not decrease, despite the simultaneous existence of the two diseases at times. Primary and/or secondary sarcopenia causes a decline in skeletal muscle performance by decreasing the number of muscle fibers and atrophy of each muscle fiber. The occurrence of falls increases by approximately three-fold in patients with sarcopenia compared with that in the same-aged population without sarcopenia [15]. Fall-induced bone fractures cause damage to skeletal muscles and cause patients to be bedridden, consequently resulting in a loss of exercise opportunities. Owing to the reduced regeneration capacity of elderly people, a single fall can accelerate sarcopenia, causing them to be bedridden. Disuse muscular atrophy often begins during hospitalization and lasts until bone fracture recovery is achieved. Disuse muscular atrophy might also be accompanied by primary sarcopenia.

Resistance training, such as squats and push-ups, is effective at preventing sarcopenia progression. However, elderly people are at risk of falling during this training. Further, the performance of a workout is difficult when bedridden. Thus, pharmacological or nutritional approaches for prevention and further recovery from sarcopenia are attractive strategies for elderly people with training difficulties.

In this review, we aimed to summarize and analyze the existing models of cell cultures, rodents, and zebrafish for skeletal muscle senescence.

## 2. Cell Culture Models for Skeletal Muscle Atrophy

Aging at the cellular level is due to mitochondrial dysfunction, the accumulation of oxidative stress, telomere shortening, and stable cell cycle arrest induced in response to intrinsic and extrinsic stimuli such as UV radiation [16,17]. Cell cycle arrest in senescent cells is unique in that it occurs in the G1 and G2 phases, in contrast to that in quiescence, which occurs in the G0 phase [18]. Unlike normal cells, senescent cells exhibit phenotypic changes that include a gradual growth arrest unresponsive to mitogenic stimuli [19,20]. This cyclic arrest is thought to be executed by the activation of p53/p21cip1 and p16INK4a/Rb, particularly in early senescence, and p53/p21cip1 and p16 INK4a/Rb activation in later stages [21,22].

For drug discovery, cell cultures are the most powerful tool, as they enable the observation of body dynamics that cannot otherwise be observed from outside the human body. By using cell cultures, many parameters can be easily measured with various quantitative techniques. Further, the physicochemical environment, such as pH, temperature, osmotic pressure, and dissolved gas concentration, can be controlled, and the physiological environment, such as hormone and nutrient concentrations and transgenes, can be modified. To reduce the use of animals due to ethical reasons, cell culture experiments are currently recommended. Cell culture studies related to skeletal muscle cell senescence can be classified as follows: the proliferation of myoblasts, differentiation from myoblasts to myotubes, functional parameters of differentiated myotubes, and cell senescence induction (Figure 1).

### 2.1. Myoblast Proliferation

The proliferation of myoblasts is expected to mimic the increase in skeletal muscle amount related to tissue regeneration, wound healing, and the recovery of muscle atrophy and sarcopenia [23]. Myoblast proliferation has been evaluated in many studies using relatively simple methods. For example, beta-hydroxy beta-methylbutyric acid, a well-known dietary supplement that enhances skeletal muscle performance with resistance training, was first demonstrated to induce myoblast proliferation using this strategy [24].

### 2.2. Myotube Differentiation

Myotube differentiation, which implies an increase in skeletal muscle amount and muscle regeneration in vivo, can be evaluated using several myoblast cell lines. For example, C2C12 mouse myoblasts and L6 rat myoblasts are well-established cell lines that can easily differentiate into myotubes [25,26]. Human skeletal muscle cells (SkMC) isolated from the skeletal muscle of a single adult donor have also been used in studies. Note that, unlike C2C12 and L6, these cells have a mitotic frequency and are not immortalized cells. SkMC may be affected by donor age and gender, and comparative parameters, such as responsiveness to dexamethasone-induced myotubular atrophy [27], may be difficult among different SkMC lines or with C2C12 cells. In addition, targeted differentiation of pluripotent stem cells into myotubes forms three-dimensional muscle constructs and is expected to go beyond conventional cell models [28].

In general, cultured myotube differentiation is quantified by immunohistological staining of proteins specific to differentiated skeletal muscle cells, such as myosin heavy chain (Mhc) protein, with multinucleate large cell shape. In addition, qPCR analysis of genes involved in skeletal muscle differentiation is an easy and reasonable approach to confirm myotube differentiation [29,30].

### 2.3. Functional Parameters of Differentiated Myotubes

Intracellular Ca^2+^ and glucose uptake are well-known biochemical parameters of differentiated myotubes. Intracellular Ca^2+^ plays an important role in the contraction of skeletal fibers involved in myosin-actin cross-bridging, fiber type shifting, etc. [31]. During normal contractions, action potentials generated by motor neurons stimulate the sarcoplasmic reticulum (SR) to release Ca^2+^ from the SR into the cytosol. Thereafter, Ca^2+^ binds to troponin C, which activates a series of contractile proteins and induces skeletal muscle contraction [32,33]. To measure intracellular Ca^2+^ levels in myoblasts and differentiated myotubes, many researchers have used Ca^2+^ indicator fluorescent dyes, such as Fura-2 [34], and fluorescent proteins, such as GCaMP [35].

Glucose uptake was used to assess the function of differentiated myotubes in vitro. Sarcopenia is prone to occur in patients with diabetes and worsens due to high blood sugar levels [36]. Improving glucose uptake in peripheral tissues, including skeletal muscle, is a promising approach to improving or preventing sarcopenia and diabetes. To measure glucose uptake in myoblasts and differentiated myotubes, fluorescent glucose analogs, such as 2-NBDG, are usually used [37]. In addition to Ca^2+^ and glucose uptake, several metabolic molecules, such as GL3P and UDP-GlcNAc [38], are also reported to be indicators of differentiated myotubes.

### 2.4. Cell Senescence Induction

The induction of cell senescence has become an attractive strategy for studying skeletal muscle senescence. D-Galactose (D-gal), dexamethasone (DEX), and TNF-α are usually used to induce cell senescence in myoblasts and differentiated myotubes. D-gal treatment increases oxidative stress and activates apoptotic pathways, which are common in conventional cell senescence [39]. For example, D-gal inhibits the proliferation of C2C12 cells and increases the expression levels of p53, which induces apoptosis, and p16, which induces cell cycle arrest [40]. In contrast, DEX induces atrogin-1, a muscle-specific F-box protein that activates the ubiquitin-proteasome pathway during muscle atrophy [41]. DEX also promotes protein degradation in L6 and C2C12 cells [42] as a mimic of protein degradation in disused muscle atrophy. Similar to DEX, TNF-α synthesizes and accumulates intracellular ceramide to inhibit myogenic differentiation [43] and promotes protein degradation in L6 and C2C12 cells [44].

Senescence-associated beta-galactosidase (SA-β-gal) activity is a major biomarker for cell senescence in almost all cell types. The activity of SA-β-gal can be easily measured and SA-β-gal can be stained using X-gal protocols [45]. Cultured human myoblasts with less ability to differentiate into myotubes exhibit strong SA-βgal activity [46], indicating that SA-βgal is an ideal biomarker for skeletal muscle senescence. Jadhav et al. also reported that the antidiabetic drug, metformin, suppresses ceramide-induced cell senescence in C2C12 myoblasts [47].

### 2.5. Limitations of Cell-Based Testing

Although the above cell culture models are quite attractive and promising for drug screening in skeletal muscle cell senescence, they are associated with several limitations owing to their artificial environments compared to animal models and clinical situations. In vivo, cells and the extracellular matrix (ECM) interact with each other, thereby affecting cell behavior, including proliferation, adhesion, migration, and differentiation [48]. Growth and development contribute to cell-ECM interactions not only in general tissues but also in normal muscles [49]. Further, cultured cells produce many reactive oxygen species (ROS) and use ROS-dependent signaling pathways, which would be non-functional in vivo [50]. Of note, the use of immortalized cell lines, such as C2C12, should be considered a different situation compared to in vivo conditions.

## 3. Rodent Models for Skeletal Muscle Senescence

As described above, several types of cell models are used to evaluate skeletal muscle cell senescence, despite the limitations associated with in vitro modeling. Thus, animal models, mainly rodents, are necessary to evaluate test compounds in skeletal muscle senescence as part of preclinical testing.

Approximately 90% of the rodent and human genomes have comparatively syntenic and orthologous genes, including myogenic regulatory factors (MRFs), with 78.5% amino acid identity [51,52]. Accordingly, rodents have been one of the most prevalent model animals for a long time. In the mouse embryo, myogenesis begins with myogenic factor 5 (Myf5) expression, which is followed by myogenin (Myog) and other MRF expression [53], similar to the process in humans. Although rodents might be suitable animal models for human skeletal muscle physiology, a difference exists in myofibril composition between rodents and humans. For example, in mice, fast muscles are widely located in the body, and slow muscles have a higher percentage in the lower limbs. In humans, although the proportions vary at different sites, slow and fast muscles are distributed in a mosaic manner [54]. Major rodent and zebrafish models used to evaluate skeletal muscle senescence are listed in Table 1.

### 3.1. Aged Models

Aging is the main risk factor for muscle atrophy and sarcopenia, and several aged rodent models have been developed using wild type and mutant strains [78]. In addition to natural aging, dietary induction has also been employed [79,80,81]; a high-fat diet is known to accelerate aging for the evaluation of anti-sarcopenic natural products [82,83].

Accelerated aging models, such as Senescence-Accelerated Mouse (SAM) mice, enable the understanding of the mechanisms of aging and skeletal muscle senescence in a relatively short experimental period. Particularly among SAM strains, senescence-prone (SAMP) mice, especially SAMP8, show features of muscle aging (reduced muscle mass, reduced tetanic contraction and relaxation rates, and atrophy of type II muscle fibers) earlier than normal mice and have more pronounced muscle aging than other SAMPs [84]. However, SAMP1, SAMP6, and SAMP10 are also used to study muscle aging, as they show fair characteristics of muscle aging [78].

There are several genetically engineered aging mice that depict senescence acceleration; for example, Lmna^L530P/L530P^ [60], Bub1b^H/H^ [62], Bub1b^+/GTTA^ [63], p53^+/m^ [64], Bmal1^−/−^ [66], and RPS9 D95N [67]. As a representative example, we describe the Lmna^L530P/L530P^ mouse. Lmna is a gene encoding lamin A, a component protein of intermediate diameter filaments that make up the nuclear lamina, and mutations in the Lmna gene are one of the causes of Hutchinson-Gilford disease. Lmna^L530P/L530P^ mice show degeneration of the cardiac and skeletal muscles, with myoblasts differentiating into adipocytes [60]. This is thought to be due to a defect in the nuclear membrane that prevents the maintenance of the chromatin structure, resulting in dedifferentiation and subsequent redifferentiation to other cell types. Since the muscles of elderly humans show fat accumulation and a similar phenotype, this mouse is considered an ideal model of skeletal muscle aging [85].

### 3.2. Sarcopenia Model

The most commonly used genetically engineered models of sarcopenia are Sod and Il-10^−/−^ mice. Mutations in copper–zinc superoxide dismutase (CuZnSOD [SOD1]) are a cause of amyotrophic lateral sclerosis (ALS) [86,87], and Sod1^G93A^ mice develop ALS [88]. In contrast, SOD1^−/−^ mice exhibit characteristics of accelerated aging as follows: increased p16 and p21 expression (mRNA and protein) [89] and consequent neuromuscular junction disruption, impaired motor nerve transmission, and accelerated muscle atrophy [90]. For this reason, SOD1^−/−^ mice have been used in various studies of muscle atrophy and sarcopenia [91,92]. Il-10^−/−^ mice are used as models of skeletal muscle aging, especially sarcopenia, because they develop age-related loss of skeletal muscle strength. Il-10 induces a switch from the M1 to the M2 phenotype of macrophages that migrate to damaged skeletal muscle; M2 macrophages are required for normal muscle growth and regeneration [93]. Il-10 has also been shown to improve insulin signaling and glucose metabolism in the skeletal muscle [94]; the rate of ATP synthesis and high-energy phosphate concentration levels are reduced in the skeletal muscle of Il-10^−/−^ mice [95].

### 3.3. Hindlimb Unloading Model

Hindlimb unloading, a technique that stimulates weightlessness and induces skeletal muscle atrophy, was developed in the mid-1970s [96]. Although hindlimb unloading was developed to study the body's response to the space environment [97], with drug screening for bone loss under microgravity as the main focus [98,99], some similarities exist between hindlimb unloading and age-induced skeletal muscle atrophy. For example, animals that received hindlimb suspensions had decreased muscle weight and a smaller fiber diameter with increased apoptosis of muscular cells via the upregulation of atrogin-1 [74], which is a common mechanism for age-related loss of skeletal muscle [100]. Yoshihara et al. demonstrated that astaxanthin supplementation ameliorated muscle atrophy induced by hindlimb unloading by inhibiting myonuclear apoptosis [101]. In addition, Ferrando et al. revealed that allopurinol prevents muscle mass loss by suppressing the expression of ubiquitin ligase [102].

### 3.4. Denervation Model

In the clinic, muscle atrophy is known to be caused by motor nerve damages. In rodents, the sciatic or tibial nerve in one leg is removed to induce denervation [103,104]; muscle atrophy is mainly caused by fast-twitch muscle fibers [105]. Notably, this model increases mitochondrial ROS production in skeletal muscle cells [76], thereby targeting the improvement of age-related loss of muscle mass and function [106,107]. However, aging might cause mitochondrial abnormalities and increased ROS production [108] (i.e., not a therapeutic target); thus, further research is necessary to address this issue.

Shen et al. revealed that isoquercitrin alleviates atrophy and mitophagy in the soleus muscle by suppressing oxidative stress and inflammatory responses. In fact, after denervation, vacuolar degeneration and autophagy were observed in the treatment group compared to the control group and were found to be accompanied by high expression of the autophagy-related proteins, ATG7, BNIP3, PINK1, and LC3B [77]. Hiramoto et al. reported that supplementation with alkylresorcinols caused the recovery of fatty acid metabolism, which occurs during muscle atrophy using this model [109].

### 3.5. D-Galactose Model

D-gal induces chronic inflammation and oxidative stress, leading to accelerated aging in rodents , which is similar to the results obtained in cell culture studies [110]. D-gal administration causes a significant decrease in the gastrocnemius muscle mass/body weight ratio and a decrease in the cross-sectional area of the skeletal muscle [71]. As this model is relatively easy to use, several studies have employed this model for drug testing. For example, dihydromyricetin (DHM) was found to alleviate the reduction in gastrocnemius weight/body weight, the cross-sectional area of skeletal muscle fibers, and fiber diameter, which deteriorated due to D-gal [72]. DHM can be expected to promote longevity by downregulating the pERK and pAKT pathways in *Drosophila* [111]. Thus, DHM may be beneficial for the treatment of skeletal muscle atrophy and its prevention during the aging process. In addition, by using a D-gal-induced sarcopenia rat model, Li et al. revealed that bovine milk fat globule epidermal growth factor VIII inhibits cell apoptosis in the gastrocnemius muscle to prevent muscle mass loss [112].

### 3.6. Dexamethasone Model

Many patients receive long-term treatment with glucocorticoids, which can induce muscle atrophy as a serious side effect [113]. DEX, a glucocorticoid medication, is known to induce proteolysis in skeletal muscles, resulting in muscle atrophy in rodents [68,114] and zebrafish [70]. Two E3 ubiquitin ligases, Atrogin-1, and MuRf-1, are upregulated in DEX-treated rodents [69,115], similar to the culture cell model [116]. These ubiquitin ligases promote ubiquitin-mediated protein degradation in skeletal muscle, which accelerates sarcopenia and is a potential target for preventing sarcopenia [117]. The increase in myostatin expression is also important in DEX-induced skeletal muscle atrophy [118]. Many drug testing studies have been performed with this model. For example, Otsuka et al. demonstrated that quercetin glycosides reduced muscle atrophy in the mouse gastrocnemius muscle by downregulating atrogin-1, MuRF-1, and myostatin, which are associated with muscle atrophy [119].

## 4. Zebrafish Models for Skeletal Muscle Senescence

Rodent organs are believed to mimic human organs, as they possess a high degree of genetic homology with human genes. Therefore, rodents are typically used as animal models for drug testing. However, remarkable time, labor, space, and money are required for studies with rodents. In recent years, from the viewpoint of animal welfare, there has been an ongoing trend to reduce the number of mammals used in experiments. An ideal alternative to rodents, the zebrafish has emerged as an animal model for several types of human diseases, including skeletal muscle atrophy [120,121], aging [122], and sarcopenia [123,124], owing to several advantages (Figure 2). The skeletal muscle constitutes a large part of the zebrafish trunk and has a high degree of similarity to human muscle, both molecularly and histologically. The skeletal muscle also consists of slow muscles located directly below the body surface and fast muscles around the vertebrae [125]. Zebrafish have a set of orthologs for human MRFs that are involved in skeletal muscle myogenesis [126]. Myod and myf5 are expressed in the early phase of skeletal muscle development, followed by myogenic factor 6 (myf6). Interestingly, myod, myf5, and myf6 expression depend on the site of the muscle tissue [127], implying that even skeletal muscle tissue might have subclassifications in zebrafish.

To induce muscle atrophy or administer drugs to rodents, intravascular or oral gavage, which requires certain techniques, must be employed. However, zebrafish can absorb chemicals from their skin and gills by simply adding them to general fish water (exposure test). Protocols have also been established for oral and intraperitoneal administration in zebrafish [128,129,130].

### 4.1. Aged Model

Aging is the main driver of skeletal muscle atrophy and sarcopenia in fish species, like in rodents [123]. However, the lifespan of mice and zebrafish is almost identical (2–3 years in laboratory conditions) [131]. As a result, there is little advantage to using wild-type zebrafish as a model of skeletal muscle aging. Owing to the ease of mutagenesis in zebrafish, various strains have been developed in recent years using genetic manipulation (using tol2 transposase), knockdown (using antisense oligonucleotides), and knockout (using CRISPR/Cas9) techniques. For example, Kishi et al. performed an SA-β-gal activity-based mutant screen and identified 11 zebrafish mutants with high SA-β-gal activity. Of these, heterozygous mutations in telomeric repeat binding factor 2 (terf2) or spinster homolog 1 (spns1) were found to result in a shorter lifespan [132]. Further, the spns1 mutation led to an increase in lipofuscin in the skeletal muscle in the adult stage, indicating that the mutant can be used as a model for skeletal muscle aging. Previously, Da Rosa et al. reported that growth hormone (GH) overexpression accelerates spinal curvature in adult zebrafish by reducing *myog* and *myod* expression in surrounding muscles [133]. Their findings highlight the possibility of accelerated skeletal muscle aging in this transgenic fish. To our knowledge, there are no reports of the use of these aged models in studies on skeletal muscle aging; however, various studies are expected in the near future.

As described above, compared to rodent models, the use of zebrafish models certainly shortens the duration of experiments and reduces the burden of animal husbandry, though it is still labor-intensive for research on aging. Since the CRISPR/Cas9 genome editing technology was reported in 2013 [134] and used in zebrafish [135], several disease models have been reported to have been built using CRISPR-mediated targeted mutagenesis (knockout). For example, the lmna gene, whose mutation is known to accelerate aging in rodents [60], is also present in zebrafish. According to a recently published report, the deletion of five base pairs (5bpΔ) in the second exon of the lmna gene by CRISPR/Cas9 genome editing resulted in skeletal muscle damage and impaired swimming [61]. The authors used this as a model of laminopathy, such as Emery–Dreifuss muscular dystrophy; however, it also has potential as a model of skeletal muscle senescence.

### 4.2. Dexamethasone Model

DEX has been used to induce skeletal muscle atrophy in zebrafish [70], like in rodents and cultured cells. Zebrafish can absorb small molecules through their skin and gills, which enables the administration of DEX by immersion in general fish water for the induction of muscle atrophy. Although limited studies have been conducted using DEX-induced zebrafish models (less than 10 papers in Web of Science), some promising studies have been reported. For example, Ryu et al. revealed that dietary supplementation with maca (*Lepidium meyenii*) induced preventive effects in DEX-induced muscular atrophy in zebrafish, with an increase in the distance traveled and speed of chasing food in the aquarium [70]. Similar to how it occurs in rodents, the mechanism of DEX-induced muscular atrophy in zebrafish is also thought to involve an increase in proteolysis due to the overexpression of atrogin-1 and murf1 ubiquitin ligase (as a part of aging-induced muscular atrophy); however, this notion has yet to be proven.

### 4.3. Chronic Alcohol Model

Prolonged and high-dose alcohol consumption induces muscle atrophy in mammals. The mechanism by which alcohol induces skeletal muscle atrophy is not fully understood; however, alcohol is known to increase the expression of ubiquitin ligase, which contributes to muscle atrophy in elderly people [117,136]. Similar to that in mammals, chronic ethanol exposure (0.5% in the general fish water for 8 weeks) can induce muscular atrophy in zebrafish [73]. Although this model has not been used for drug testing, it could serve as a useful screen for muscle atrophy in response to alcoholism in the future.

## 5. Zebrafish Models—Future Perspective

Previous research findings on skeletal muscle senescence have been largely based on cell cultures and rodent models. During the past decade, zebrafish have attracted attention as a model vertebrate for human diseases, although the contribution of zebrafish models to the study of skeletal muscle senescence has been relatively small. However, with the aforementioned rise of CRISPR/Cas9 technology, the number of possible models of skeletal muscle senescence has been increasing in recent years. For example, in addition to the lmna crispant described above [61], tp53 crispants have overall reduced activity and shorter migration distances [65]; Tp53 is well known for its tumor suppressor activity. Therefore, senescence is reported to be accelerated in Tp53^−/m^ mice [64], and the knockout of tp53 in zebrafish is predicted to partially affect the skeletal muscle aging pathway. A model of aging due to metabolic abnormalities, such as the muscle atrophy zebrafish model of insulin receptor deficiency, which has already been reported in rodents, may also promote such research [137]. In addition, as the subset of muscle- or aging-related genes, including atrogin-1 [138] and murf1 [139], are common to humans and zebrafish, various zebrafish models are expected to emerge in the near future to stimulate research on skeletal muscle aging.

The use of zebrafish models will enable the discovery of new target genes involved in skeletal muscle senescence. For example, the knockout of foxm1, a master regulator of aging-associated cell senescence that is involved in cell cycle regulation, was found to increase myofiber death and clearance [140]. It is expected that the screening of compounds that reduce skeletal muscle senescence and the search for therapeutic target genes using zebrafish models will become mainstream in the near future.

## Figures and Tables

**Figure 1 molecules-27-08625-f001:**
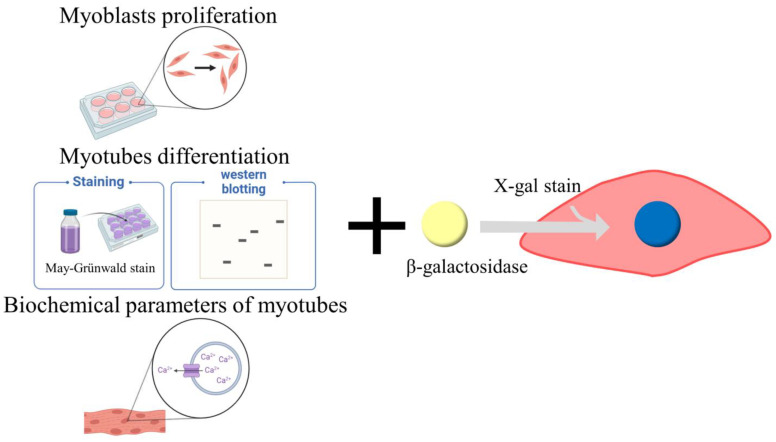
Cell-based evaluation for skeletal muscle cell senescence. Created using BioRender.com (accessed on 10 September 2022).

**Figure 2 molecules-27-08625-f002:**
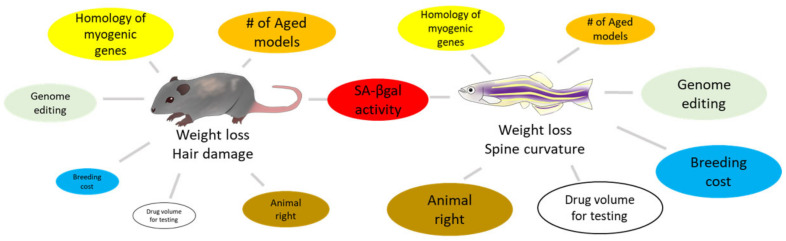
Comparison between rodents and zebrafish for skeletal muscle senescence. The size of each circle visualizes its advantage as a model animal.

**Table 1 molecules-27-08625-t001:** Rodent and zebrafish models for skeletal muscle senescence.

Type		Model	Phenotypes/Detection in Skeletal Muscle	Reference
Aged model	Natural aging	C57BL mouse (>18 m) Zebrafish (>25 m)	Sarcopenic changes Decline in physical activity	[55] [56]
	Accelerated aging	SAMP8 mouse	Sarcopenic changes	[57]
Gene knockout		SOD1^−/−^ mouse	Loss of muscle Degeneration of neuromuscular junctions Increase in muscle mitochondrial ROS	[58]
		Lmna^L530P/L530P^ mouse Lmna^−/−^ zebrafish	Reduction of muscle fiber widths Degeneration of skeletal muscle	[59,60] [61]
		Bub1b^H/H^ mouse	Muscle atrophy	[62]
		Bub1b^+/GTTA^ mouse	Reduction of muscle fiber widths Early decline in physical activity	[63]
		p53^+/m^ mouse p53^−/−^ zebrafish	Loss of muscle Muscle atrophy Reduction of activity	[64] [65]
		Bmal^−/−^ mouse	Loss of muscle Reduction of muscle fiber widths	[66]
Gene knockin		RPS9 D95N mouse	Early decline in physical activity	[67]
Chemical induced	Dexamethasone	Mouse zebrafish	Loss of muscle Upregulation of atrogin-1, Murf-1 Reduction of muscle fiber widths	[68,69] [70]
	D-galactose	Mouse	Reduction of muscle fiber widths Upregulation of atrogin-1, Murf-1	[71,72]
	Alcohol	zebrafish	Reduction of muscle fiber widths	[73]
Hindlimb unloading		Mouse	Loss of muscle Reduction of muscle fiber widths Upregulation of atrogin-1, Murf-1	[74,75]
Denervation		Mouse	Loss of muscle Reduction of muscle fiber widths Increase in muscle mitochondrial ROS	[76,77]

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
