# Peer review of "Zebrafish Models for Skeletal Muscle Senescence: Lessons from Cell Cultures and Rodent Models"

_molecules, 2022, doi:10.3390/molecules27238625_

Round 1

Reviewer 1 Report

This review describes the past studies of muscle cell senescence and muscle aging. it is well written with an interesting comparative insight of mice and zebrafish. This reviewer supports the acceptance of this manuscript in Molecules as in the current version.

Author Response

General response

We highly appreciate the time and efforts of the editor and reviewers to understand the manuscript thoroughly. Their constructive feedback has provided better insight and the reachability of this review. We assure that all the suggestions have been analyzed, and incorporated into the manuscript carefully. Response to reviewers and changes in the manuscript are in blue and red font color, respectively. In addition, we have made minor changes in the other text according to the professional English proofreading (2nd round). We are hopeful that the revised manuscript is now in better form and can be accepted for publication in Molecules.

Reviewer 1’s comment

This review describes the past studies of muscle cell senescence and muscle aging. it is well written with an interesting comparative insight of mice and zebrafish. This reviewer supports the acceptance of this manuscript in Molecules as in the current version.

Authors’ response:

We would like to thank Reviewer 1 for your review.

Reviewer 2 Report

In this review, the authors aim to summarize models for skeletal muscle senescence.  In their introduction section, they pointed out the significance of studying skeletal muscle maintenance in aging people, as well as two diseases in elderly people: disuse muscular atrophy and sarcopenia. They then reviewed current cell models, rodent models, and zebrafish models for studying skeletal senescence. Because skeletal muscle senescence is a relatively new topic with obvious clinical relevance, a review on this topic is timely. However, as detailed below, there are several concerns on the quality of this review.

Major Concerns

1.      While senescence is one of the keywords of the review, many paragraphs are not directly relevant to senescence.  The senescence part should be written in more details. For example, it is suggested to discuss key signaling pathways for senescence for each type of rodent models.

2.      In both abstract and the last sentence of the introduction section (Line 21 and 69), it appears that the authors promise to review the current status of drug discovery.  However, the manuscript deviates from this goal - there is very little information on this topic.  Please either remove these statements or add more information to review current status of drug discovery using these models, the precise pathways and molecules that have been identified for therapeutic benefits.

3.      The scope of the models needs to be expanded.  For example, in cell culture models, other cell types besides C2C12 that have been used for skeletal muscle research need to be mentioned (Ex, SkMC). In the rodent models, additional models with that are related to muscle disease/disorders and aging need to be added, such as Ex. Rps9 D95N, Progeria syndrome mouse Bub1bH/H, Bub1b+/GTTA.

4. The conclusion section is poorly presented.  It is suggested to mention the future perspectives and how to translate the current discoveries to clinical trial, if there is any.

Minor concerns

1. The keywords should be written using MeSH terms.

2. The key points of this review need to be mentioned in the abstract section.

3. In the introduction, paragraph one, line 31, a precise reference to the average life expectancy of different countries is required.

4. Please include updated references for parts 2.1, 2.2, and 2.3.

2.1 ex: Gerrard, Jeffrey C., et al. "Current Thoughts of Notch’s Role in Myoblast Regulation and Muscle-Associated Disease." International Journal of Environmental Research and Public Health 18.23 (2021): 12558.;

2.2 ex: Selvaraj, Sridhar, et al. "Screening identifies small molecules that enhance the maturation of human pluripotent stem cell-derived myotubes." Elife 8 (2019): e47970.

2.3 ex: Jang, Mi, et al. "Serum-free cultures of C2C12 cells show different muscle phenotypes which can be estimated by metabolic profiling." Scientific reports 12.1 (2022): 1-15.

5        Senescence quantification/determination should be improved by summarizing the most common senescence techniques and markers.

6. More Tables (Ex, senescence markers in muscle diseases) and images (Ex, Disuse muscular atrophy and Sarcopenia) would help with the understanding of the review.

Reviewer 3 Report

The review from Ichii et al. highlight the use of zebrafish as a model for studying skeletal muscle senescence. The authors comprehensively describe the various in vitro, rodent and zebrafish models with a goal of comparing the advantages of these. The review is well written and easy to follow however there are some areas that can be expanded for clarity. One criticism is the title focuses on Zebrafish as a model for skeletal senescence, but a large part of the review is focused on rodent and cell culture models. More lines of text refer to these models than to zebrafish itself, so it is not clear how this review is focused on the fish.

Major points:

1)    Line 173, SAMP18 is introduced as a model for muscle aging in mice, but there is no other description as to what this gene is and why this is a good model. The authors should expand on this and provide more information on this gene. 

2)    This is also the case for IL-10 and sod1 mutant mice outlined in Lines 175. The authors can outline the assays that were performed and how this led to their conclusion that these mutant mice are excellent to model muscle aging and atrophy. 

3)    It is not clear what “breeding water” means in line 257 and 287. Is this not general fish water?

4)    The authors could expand on what they mean by genetic manipulation and its ease in zebrafish for generating muscle aging models. The approach they refer to involves the generation of mutations using a chemical mutagen and screening for fish with phenotypes of high SA-b-activity (lines 268-272). Such an approach is labor intensive as compared to targeting genes using CRISPR/Cas9. 

5)    The authors speculate that DEX treatment in zebrafish causes muscle atrophy via induction of atrogin-1 expression. While they say this is expected, it may be better to further tone down this as it is a prediction or hypothesis waiting to be tested (See Lines 293-296.) The authors also mention that murf1 has not been identified in zebrafish (line 296). This is not true as it is listed as trim63a and there is a publication that describe its function in the heart. The gene is listed on ZFIN with an expression pattern in the somitic muscle and the heart. See: http://zfin.org/ZDB-GENE-040625-139#summary and Shimizu et. al. (The Calcineurin-FoxO-MuRF1 signaling pathway regulates myofibril integrity in cardiomyocytes eLife 6:e27955. 2017).

6)    The authors could expand on the use of zebrafish as a model for drug screening in relation to muscle atrophy. This is stated in the abstract and introduction, but there is not text that presents this case. I am not entirely convinced that zebrafish is an excellent model for drug screening with respect to muscle aging. 

7)    Missing references: A total of 104 references are cited in the manuscript, but only 98 are listed in the bibliography. 

Round 2

Reviewer 3 Report

In the revised version, the authors have addressed all of my comments. This is a great review on the comparison of various model organisms and cell culture studies on senescence.